# Endorsement of COVID-19 misinformation among criminal legal involved individuals in the United States: Prevalence and relationship with information sources

Xiaoquan Zhao[1]*, Aayushi Hingle[2], Cameron C. Shaw[3], Amy Murphy[3], Breonna R. Riddick[1], Rochelle R. Davidson Mhonde[4], Bruce G. Taylor[5], Phoebe A. Lamuda[5], Harold A. Pollack[6], John A. Schneider[6,7], Faye S. Taxman[3]

1 Department of Communication, George Mason University, Fairfax, Virginia, United States of America, 2 Department of ELAP, Linguistics, & Communication Studies, Montgomery College, Takoma Park, Maryland, United States of America, 3 Schar School of Public Policy, George Mason University, Fairfax, Virginia, United States of America, 4 College of Public Health, George Mason University, Fairfax, Virginia, United States of America, 5 NORC at the University of Chicago, Chicago, Illinois, United States of America, 6 Crown Family School of Social Work, Policy, and Practice, University of Chicago, Chicago, Illinois, United States of America, 7 Department of Medicine, University of Chicago, Chicago, Illinois, United States of America

* xzhao3@gmu.edu

## Abstract

Criminal legal system involvement (CLI) is a critical social determinant of health that lies at the intersection of multiple sources of health disparities. The COVID-19 pandemic exacerbates many of these disparities, and specific vulnerabilities faced by the CLI population. This study investigated the prevalence of COVID-19-related misinformation, as well as its relationship with COVID-19 information sources used among Americans experiencing CLI. A nationally representative sample of American adults aged 18+ (N = 1,161), including a subsample of CLI individuals (n = 168), were surveyed in February-March 2021. On a 10-item test, CLI participants endorsed a greater number of misinformation statements (M = 1.88 vs. 1.27) than non-CLI participants, p < .001. CLI participants reported less use of government and scientific sources (p = .017) and less use of personal sources (p = .003) for COVID-19 information than non-CLI participants. Poisson models showed that use of government and scientific sources was negatively associated with misinformation endorsement for non-CLI participants (IRR = .841, p < .001), but not for CLI participants (IRR = .957, p = .619). These findings suggest that building and leveraging trust in important information sources are critical to the containment and mitigation of COVID-19-related misinformation in the CLI population.

## Introduction

Misinformation poses a significant threat to effective public health responses to the COVID-19 pandemic [1–3]. Ample research has focused on the prevalence, impact, and correction of

**Data Availability Statement:** All relevant data are within the manuscript and its Supporting Information files.

**Funding:** This work was supported by funding (U2CDA050097, PI: FST; U2CDA050098, PIs: HP and JS) from the U.S. National Institute of Health's (NIH) National Institute on Drug Abuse (NIDA), Justice Community Opioid Innovation Network (JCOIN), protocol approval number JCOIN 026. U2CDA050098 provided a subcontract, AWD100228, with the University of Chicago. The views expressed do not necessarily reflect the policies of NIDA. NIDA staff worked with the study investigators on the study research questions, measures and research design, but had no involvement in the collection, analysis, or interpretation of data, in the writing of this report, or in the decision to submit this article for publication. There was no additional external funding received for this study.

**Competing interests:** The authors have declared that no competing interests exist.

COVID-19 misinformation in the general population [4, 5]. Less attention has been paid to misinformation among minoritized and marginalized communities [1]. No research to date has specifically examined the presence and dissemination of misinformation among individuals with criminal legal system involvement (CLI), a population that has traditionally faced significant health disparities. In this study, we assess the prevalence of COVID-19 misinformation among a nationally representative sample of CLI and non-CLI individuals in the United States (U.S.). As the emergence, spread, and mitigation of misinformation are often observed in specific information channels, the associations between sources used for COVID-19 information and misinformation endorsement within CLI vs. non-CLI groups are also investigated.

## CLI as a social determinant of health

CLI refers to current or prior experience of involvement in the criminal legal system such as arrest, incarceration, probation or parole, or another legal status. According to Healthy People 2030, social determinants of health are "the conditions in the environments where people are born, live, learn, work, play, worship, and age that affect a wide range of health, functions, and quality-of life outcomes and risk" [6]. For many CLI individuals, their history of involvement in the criminal legal system including prison and jails constitutes an important condition that profoundly shapes who they are, how they live their lives, and how they pursue health and well-being. As a unique social determinant of health, CLI is intertwined with other structural influences on health, such as racial discrimination, poverty, food and housing insecurity, education quality, health care access, and neighborhood environments [6]. CLI can impact health directly via the conditions of legal involvement; it may also impact health indirectly through CLI's connection to other social determinants of health, such as economic well-being. Adults on probation, for example, often face difficulties in securing employment, food, and housing, which places them at high risk for both recidivism and adverse health outcomes [7].

Individuals with a history of incarceration experience worse health outcomes compared to the general population. Incarcerated people are more likely to have high blood pressure, asthma, cancer, arthritis, and infectious diseases [8]. Half of all people in state and federal prisons have a chronic medical condition [9]. More than half of individuals with opioid use disorder have had contact with the criminal legal system and as the level of opioid use increases, involvement in the justice system also increases [10]. More than one-third (37% to 44%) of incarcerated individuals report ever being diagnosed with a mental health disorder [11]. Roughly one in four women and one in ten men in the criminal legal system have co-occurring substance use disorders (SUDs) and mental health disorders [12–14]. Using thirteen years of data from the National Survey on Drug Use and Health (NSDUH, 2002–2014), Fearn and colleagues showed that individuals who self-disclosed being on probation or parole were four to nine times more likely than their non-CLI counterparts to report SUDs and that these disparities had changed little over the study period [15].

The recognition of CLI as a social determinant of health compels effort to mitigate its negative impact. Intervention strategies such as drug treatment courts and enhancing access to comprehensive health care during and after incarceration have potential to protect and improve the well-being of the CLI communities. However, the current evidence base lacks attention to health information access and consumption by CLI individuals where systematic differences between CLI and non-CLI communities may have contributed to differential health outcomes. This study addresses this gap by examining the prevalence of health misinformation among the CLI population during the COVID-19 pandemic. Evidence on differential vulnerability to COVID-19 misinformation may provide important insights to inform future efforts to address CLI-related disparities during health emergencies and beyond.

## CLI disparities during the pandemic

Health disparities faced by the CLI population place individuals at heightened risk for both infection and death during the COVID-19 pandemic. Correctional environments present a number of factors that increase risk of COVID-19 exposure to both residents and staff [16], including an aging incarcerated population with high rates of underlying and chronic health conditions; overpopulated, confined spaces and unsanitary conditions; limited healthcare capacity; as well as continuous movement within, into, and out of the facilities by residents, staff, attorneys, and other visitors [16–19]. The UCLA COVID-19 project shows that, as of the current writing, at least 663,196 COVID-19 cases among incarcerated individuals have been reported, including 3,181 deaths, plus an additional 247,194 cases and 311 deaths among carceral institution staff [20]. Data from early in the pandemic indicated that incarcerated individuals were 5.5 times more likely to be infected with COVID-19 and 3 times more likely to die from it than the rest of the U.S. population [21].

The disproportionate effect of incarceration on marginalized communities defined by race, ethnicity, and socioeconomic status is well established [17, 22, 23]. For Black, Indigenous, and other people of color (BIPOC), systemic barriers to equitable healthcare are a continuous concern [24–26]. These disparities are reflected in higher rates of COVID-19 hospitalizations and mortality among BIPOC communities [27–29]. In general, BIPOC communities are 2.5 times more likely to be hospitalized and 1.7 times more likely to die from COVID-19 than White individuals. Hospitalization and death rates are slightly higher for Black individuals than Hispanic or Latinx individuals, and Native or Indigenous communities face the highest rates of infection and death [30]. The complex nexus of racial, socioeconomic, and CLI disparities poses a unique challenge to health equity during the pandemic.

In analyzing COVID-19 protective behaviors among various populations, Schneider et al. [31] indicate that CLI individuals are overall less likely to report protective behaviors, which may be a result of employment factors, limited access to personal protective equipment, and crowded housing or homelessness. Notably, CLI individuals are less likely to use face coverings but more likely to be tested than other populations of the study. The authors note that testing behaviors may be a direct effect of involuntary confinement in crowded carceral settings or needed for access to community services. This suggests that the CLI population's responses to the COVID-19 pandemic are profoundly shaped by the social, environmental, and legal factors in their lived experiences.

## COVID-19 misinformation

Misinformation about COVID-19 is so widespread and influential that the World Health Organization (WHO) declared that the pandemic is concurrently an "infodemic" [32]. Misinformation is defined as "information that is false, inaccurate, or misleading according to the best available evidence at the time" [1]. Some consider misinformation a meta-risk during the COVID-19 pandemic as it interacts with and complicates perceptions about the original risk [33]. The threat of a deadly virus, coupled with evolving uncertainty and increasing politicization of the pandemic, has fueled rampant growth and dissemination of misinformation. False information tends to diffuse "farther, faster, deeper, and more broadly" than accurate information [34]. An early assessment during the pandemic reported that online messages from medical and public health authorities, such as the WHO and Centers for Disease Control (CDC), generated much less public engagement than platforms hosting misinformation and conspiracy theories [35]. There is evidence that this pattern has persisted as the pandemic continues [36].

Prevalent misinformation can influence public perceptions and pandemic response, with strong documented associations between COVID-19 misinformation exposure and maladaptive knowledge, beliefs, and behaviors [3, 37–39]. A study in Hong Kong found that exposure to misinformation about smoking and drinking being protective factors against COVID-19 was associated with increased use of both substances among current users [40]. In the U.S., endorsing misinformation about COVID-19 was found to undermine prevention self-efficacy, which could in turn negatively impact preventive behaviors [41]. Experimental data from Ireland showed that a single exposure to fabricated news stories about COVID-19 could generate measurable effects on protection-relevant behaviors [37]. Given the critical importance of vaccination to combat against COVID-19, misinformation about the vaccines is particularly detrimental [42]. A recent randomized controlled trial found that exposure to misinformation could decrease vaccination intent by 6.2% in the U.K. and 6.4% in the U.S. among those who had previously indicated intention to get vaccinated [2].

Recent data from the COVID States Project indicate that belief in vaccine misinformation is intricately related to sociodemographic variables in the U.S. [3]. Men, 35 to 44 -year-olds, African Americans, Hispanic or Latinx/e people, parents of young children, those with lower education, and Republicans are more likely to hold vaccine misperceptions. Moreover, the relationships between misinformation acceptance and socioeconomics appear to have evolved over time. For example, those with high education and income were among the most likely to accept vaccine myths early in the pandemic. The same groups, however, are now among the least likely to endorse vaccine misinformation [3].

Although some sociodemographic differences in misinformation have varied across studies [cf. 41], the partisan divide on COVID-19 misinformation acceptance has been demonstrated repeatedly [41, 43, 44]. In general, individuals leaning Republican are more likely to hold COVID-19 misperceptions than those leaning Democrat. Furthermore, the ability to withstand exposure to misinformation appears to vary across gender, income, and racial and ethnic groups [2]. No research to date has examined the prevalence of COVID-19 misinformation in the CLI population, although studies of general COVID-19 knowledge and beliefs in this population point to the possibility of increased vulnerability [45, 46].

## COVID-19 information sources

The COVID-19 pandemic happened at a time of unprecedented technological advancement and fragmentation in the informational environment [47]. A long research tradition in health communication focuses on the relationship between the usage of different information sources and health attitudes and behaviors [48–50]. The plethora of information sources available today suggests that differential reliance and use of sources may lead to distinctive patterns of information gain, including the reception and acceptance of misinformation, by different populations. One analysis of fact-checked misinformation in 2020 showed that the vast majority of the identified misinformation appeared on social media (88%), followed by TV (9%), published news outlets (8%), and other websites (7%) [51]. However, these findings have to be interpreted with caution because the sampled misinformation represented a particularly influential category (i.e., false ideas widespread enough to catch the attention of fact-checking organizations). Moreover, high prevalence of misinformation on social media does not mean that the net effect of social media use is necessarily detrimental because social media also include critical outlets for accurate health information. On this latter point, there is research showing that participants in the U.S. receiving more COVID-19 information online report more frequent efforts to engage in all types of preventive behaviors [52], although counter evidence also exists [44].

A few studies have examined the relationship between media source usage and COVID-19 misinformation endorsement. A national probability survey conducted in the U.S. in early 2020 showed low perceived lethality of the coronavirus and high endorsement of misinformation [44]. Moreover, exposure to mainstream broadcast and print media was correlated with accurate risk perceptions and less belief in misinformation, whereas exposure to conservative media (e.g., Fox News) was correlated with higher levels of misinformation. Another nationally representative survey conducted a few months later replicated the findings on the different associations between partisan media use and misinformation endorsement [41]. It showed that people relying on conservative media sources tended to score higher on COVID-19 misinformation. In contrast, those using liberal media (e.g., MSNBC), mainstream print, or social media as primary sources of COVID-19 news tended to score lower. Additional studies have produced mixed findings regarding specific information sources. A multinational study, for example, found that exposure to traditional media (e.g., television, radio, newspapers) was associated with lower belief in COVID-19 conspiracy theories and misinformation [53]. Conversely, another study done in the U.S. found that higher news consumption through traditional media was associated with lower knowledge and more fake news beliefs [54].

It is important to note that trust is a critical determinant of the types of information CLI individuals seek out for health and how information consumption may alleviate or deepen disparities depending on the information obtained [55, 56]. Trust is often determined by the intersections of personal and social identities such as race, ethnicity, socioeconomic status, and region, to name a few [57–59]. Individuals involved in the criminal legal system often turn to supplemental or nontraditional sources, such as peer navigators, for health information because of general distrust in and negative experiences with the government and medical establishment [60, 61]. A recent study of CLI women found that Black women chose TV as their most trusted source of information regarding COVID-19, while White women chose government or social service agencies as their most trusted sources [62]. Additionally, 15% of the women studied reported not trusting any sources of information. Trust, or the lack thereof, factors significantly in information consumption among the CLI population during the pandemic. For racially minoritized groups in this population, the pattern and level of trust in information sources most likely reflect the dual influence of structural racism and historical mistreatment [63].

## Current study

Although COVID-19 misinformation has garnered significant attention during the pandemic, research has only begun to examine this phenomenon from a health equity perspective. This study focuses on COVID-19 misinformation among a general population inclusive of a unique high-risk subpopulation–individuals with criminal legal system involvement. As a social determinant of health, CLI has been linked to an array of health disparities, but little attention has been paid to how the CLI communities have been affected by unequal burdens of health misinformation. Our first aim is thus to assess the prevalence of misinformation among CLI individuals as compared to non-CLI individuals. Our second aim is to assess the relationship between misinformation endorsement and patterns of information source usage within each of these two groups. The overarching goal is to shed light on misinformation as a unique form of disparity affecting CLI communities during the pandemic and the extent to which use of information sources may be associated with this disparity.

RQ1: Does the prevalence of COVID-19 misinformation vary between CLI and non-CLI individuals?

RQ2: How is COVID-19 misinformation endorsement associated with information source usage among CLI and non-CLI individuals, respectively?

## Method

### Survey

Data used in the study came from the AmeriSpeak Omnibus survey conducted by NORC at the University of Chicago. AmeriSpeak is a probability-based panel of about 35,000 households recruited using area probability and address-based sampling. The panel provides sample coverage of approximately 97% of the U.S. household population. The AmeriSpeak Omnibus survey is conducted monthly with a nationally representative sample of adults aged 18 and older drawn from the AmeriSpeak panel with probability sampling based on sex, age, race/ethnicity, and education. Most AmeriSpeak households participate in surveys online through either conventional internet connection or smartphone access, and non-internet households can participate in AmeriSpeak surveys by telephone. The questionnaire used for this study was fielded in February and March of 2021 (N = 1,161). Survey invitations were sent out by email to the selected panel members. Those who did not respond to the initial invitation were contacted multiple times by email and phone. The survey was offered in both English and Spanish. Respondents received an incentive worth $4 for their participation. The overall response rate for this survey was 11.1% (37% panel recruitment response rate multiplied by 30% within-panel study-specific response rate).

AmeriSpeak's recruitment procedures for protecting the rights of human research subjects have been reviewed and approved by NORC's IRB. NORC obtains and documents research subjects' informed consent for panel participation and agreement to the study's Privacy Policy and Terms and Conditions either online or over the phone during the registration process. Upon completion of the registration process and an introduction survey, respondents become "active" AmeriSpeak panelists eligible for client studies and additional NORC-conducted profile surveys. For the current study, a key information statement was included in the survey on the topics covered and the time requirement for completion. A request for alteration of consent and waiver of documentation of consent for this study was submitted to and approved by NORC's IRB. Through this process, NORC's IRB granted full approval to the study described in this manuscript.

### Measures

The survey covered a wide range of questions, including a module specifically designed for this study. In this module, respondents were presented with 10 statements about COVID-19 (six false and four true) and asked to indicate each as true or false (see Table 3 for items). The statements were all based on misinformation extracted from earlier focus group research with a sample of CLI individuals recruited from several central states in the U.S. Methodological details and findings from the focus group research are presented elsewhere [64]. To minimize straight-lining in responses, several of the misinformation items were reworded to be factual/accurate in nature (e.g., "I cannot get COVID-19 by getting tested for it"). Responses to factual statements were reverse coded to be consistent with the coding of misinformation statements. The total number of incorrect responses across the 10 items was tallied as a measure of overall misinformation endorsement.

Respondents reported the sources from which they obtained most of their information about COVID-19 in the past month (yes or no). Fifteen sources were shown and we grouped the sources based on joint consideration of (a) the nature of each source, (b) commonly used

categorization schemes in previous research [e.g., 41, 44], and (c) an exploratory factor analysis using Mplus (v.8) and weighted least squares mean and variance estimator (WLSMV) given the categorical data. In the end, five source categories were created: (a) government and science: federal government, local and state government, and scientific journals; (b) mainstream and news media: broadcast TV, cable TV, national print, local print, radio news, and online news; (c) social media; (d) personal sources: personal networks and employer; and (e) community sources: community organizations, church, and other. An index for each source category was created by adding positive responses (yes) within the category, and the indexes correlated mildly with one another (max. r = .414).

To measure criminal legal system involvement (CLI), the survey asked whether the respondent had ever been "convicted of any misdemeanor or felony crime" or "been incarcerated in jail or prison." A positive response to either indicated CLI.

Demographics and other background variables included biological sex, age, race/ethnicity, education, income, marital status, employment status, and political party identification. Most of these variables were gathered by the AmeriSpeak panel and updated annually. Employment status and political party identification were asked in the current survey to ensure the information was most up to date.

## Analysis

Descriptive analysis was performed on all study variables both for the full sample and for sub-samples stratified by CLI. We used chi-square tests to compare CLI and non-CLI participants on sample characteristics and information source usage. For misinformation endorsement, we employed logistic regression for individual items and Poisson regression for the total number of misinformation statements endorsed. We ran two sets of Poisson models, one for the full sample, the other for CLI and non-CLI subsamples. The full sample models included CLI status, source usage, and their interactions (one interaction at a time to minimize the threat of multicollinearity), in addition to demographic controls. The interaction terms were constructed using mean-centered source usage variables. The subsample models (stratified by CLI status) only included source usage variables and demographic controls. Model building was mindful of the modest sample size for CLI individuals as compared to their non-CLI counterparts. The number of covariates in the models was restricted to ensure that at least 10 cases were available for each predictor [65, 66], even though recent methodological literature suggests that even lower cases to predictor ratios are acceptable [67, 68]. Unless otherwise noted, all analyses were weighted to align with national benchmarks and account for selection probabilities and non-response in sampling. Missing data were minimal (max = 1.1% in any analysis) and handled by listwise deletion. Multicollinearity diagnostics did not reveal any cause for concern in any of the regression models. The software package used for analyses was IBM SPSS v.28.

## Results

### Sample characteristics

Sample characteristics are summarized in Table 1. Compared to non-CLI participants, CLI participants were more likely to be male, 30 to 59 years in age, and non-Hispanic Black; more likely to report less than college education and the lowest level of income; less likely to be currently married; less likely to be Democrats and more likely to be Independent or neutral in political leaning. Given that the AmeriSpeak survey used probability-based sampling, the CLI subsample was likely to be representative of the larger CLI population. But it is difficult to evaluate the subsample's representativeness in precise terms because no national statistics on the CLI population are currently available.

**Table 1. Sample characteristics.**

| | Total Unweighted % N = 1,161 | Non-CLI Weighted % N = 989 | CLI Weighted % N = 168 | p |
|---|---|---|---|---|
| Sex | | | | .009 |
| Male | 47.8 | 46.7 | 57.5 | |
| Female | 52.2 | 53.3 | 42.3 | |
| Age | | | | .016 |
| 18–29 | 16.7 | 21.6 | 15.7 | |
| 30–34 | 30.4 | 23.8 | 31.3 | |
| 45–59 | 23.5 | 23.6 | 29.1 | |
| 60+ | 29.4 | 31 | 23.9 | |
| Race/Ethnicity | | | | .045 |
| NH White | 63.4 | 64.0 | 54.7 | |
| NH Black | 13.2 | 11.2 | 16.6 | |
| Hispanic | 16.9 | 16.7 | 16.9 | |
| Other | 6.5 | 8.1 | 11.8 | |
| Education | | | | < .001 |
| Less than HS | 4.8 | 9.5 | 11.8 | |
| HS graduate | 16.8 | 25.7 | 39.9 | |
| Some college | 43.9 | 27.9 | 26.6 | |
| Bachelor's degree | 19.6 | 20.9 | 13.8 | |
| Graduate degree | 14.8 | 16.1 | 8.0 | |
| Income | | | | < .001 |
| < $30k | 22.9 | 22.2 | 36.2 | |
| $30K - <$60K | 28.6 | 29.5 | 23.2 | |
| $60K - <$100K | 26.6 | 23.8 | 27.8 | |
| $100k+ | 21.9 | 24.6 | 12.8 | |
| Marital status | | | | < .001 |
| Currently married | 48.2 | 50.1 | 37.0 | |
| Never married | 24.9 | 25.6 | 26.1 | |
| Divorced | 11.4 | 10.0 | 18.9 | |
| Other | 15.5 | 14.3 | 18.0 | |
| Employment status | | | | .206 |
| Currently employed | 63.9 | 58.0 | 63.4 | |
| Other | 36.1 | 42.0 | 36.6 | |
| Political party/leaning | | | | .018 |
| Democrat | 37.8 | 35.3 | 28.0 | |
| Lean Democrat | 10.6 | 11.2 | 9.1 | |
| Independent/None | 15 | 14.2 | 23.5 | |
| Lean Republican | 10.1 | 10.5 | 12.4 | |
| Republican | 26.0 | 28.4 | 26.9 | |

CLI = criminal legal involved. P-values were from chi-square tests comparing the CLI and non-CLI groups.

## Misinformation endorsement

To answer RQ1, endorsement levels for the 10 misinformation statements among CLI and non-CLI participants are presented in Table 2. About three quarters of the CLI participants (72.6%) and 66.5% of non-CLI participants endorsed at least one misinformation statement, a difference that did not reach statistical significance, $\chi^2 = 2.448$, df = 1, p = .130. The average number of

**Table 2. Endorsement of misinformation.**

| Statement | % Endorsing Non-CLI | % Endorsing CLI | Un-adj. OR (95% CI) | Adj. OR (95% CI) |
|---|---|---|---|---|
| People with certain blood types will never get COVID-19. | 4.6 | 9.6 | 2.218 (1.224, 4.019) | 2.069 (1.109, 3.861) |
| African Americans are less likely to get COVID-19 compared to other racial groups. | 6.8 | 5.5 | .798 (.394, 1.617) | .746 (.360, 1.547) |
| A vaccine for COVID-19 is already available. (R) | 6.1 | 14.1 | 2.497 (1.505, 4.143) | 2.363 (1.385, 4.032) |
| I cannot get COVID-19 by getting tested for it. (R) | 35.4 | 44.1 | 1.406 (1.024, 1.987) | 1.075 (.752, 1.537) |
| If I eat right, exercise, and take good care of my body, I don't need to worry about getting COVID-19. | 7.9 | 12.8 | 1.702 (1.025, 2.826) | 1.488 (.871, 2.543) |
| COVID-19 is a scheme for rich people and big companies to make money off of the testing and vaccines. | 13.4 | 21.9 | 1.804 (1.198, 2.717) | 1.690 (1.084, 2.636) |
| The COVID-19 vaccines are coming out so fast because they have not been carefully tested. | 33.7 | 52.2 | 2.917 (1.575, 3.064) | 2.131 (1.486, 3.054) |
| Taking HIV/AIDS medications would protect me from COVID-19. | 2.2 | 5.3 | 2.440 (1.099, 5.415) | 1.844 (.798, 4.445) |
| A person with COVID-19 who wears a mask can still spread COVID-19 to other people. (R) | 11.5 | 12.1 | 1.065 (.644, 1.759) | .812 (.479, 1.376) |
| If someone I live with has COVID-19, it increases my chance of getting COVID-19. (R) | 5.4 | 10.2 | 1.984 (1.121, 3.512) | 1.659 (.911, 3.018) |
|  | **M (SD)** | **M (SD)** | **Un-adj. IRR** | **Adj. IRR (95% CI)** |
| Total # of statement endorsed | 1.27 (1.34) | 1.88 (1.87) | 1.475 (1.298, 1.676) | 1.299 (1.139, 1.481) |

CLI = criminal legal involved. OR = odds ratio. IRR = incidence rate ratio. CI = confidence interval. Adj. = adjusted. R = reverse coded. Adjusted OR controlled for gender, age, race/ethnicity, education, income, marital status, employment status, and political party/leaning.

misinformation statements endorsed by CLI participants was 1.88, compared to 1.27 by non-CLI participants. This difference was statistically significant, t = 5.123, df = 1,155, p < .001, d = .427.

For most statements, the endorsement rate was relatively low in both subsamples (around or below 15%). Two statements, however, received strong endorsement from both CLI and non-CLI participants: "I cannot get COVID-19 by getting tested for it" (reverse coded; 44.1% for CLI and 35.4% for non-CLI) and "The COVID-19 vaccines are coming out so fast because they have not been carefully tested" (52.2% for CLI and 33.7% for non-CLI). The endorsement level was significantly higher among CLI participants than among non-CLI participants for 8 statements in bivariate logistic regression (see un-adj. ORs in Table 2). In multivariable logistic regression controlling for an array of demographic, socioeconomic, and political background variables, four of the differences remained significant (see adj. ORs in Table 2). These included: "People with certain blood types will never get COVID-19;" "A vaccine for COVID-19 is already available (reverse coded);" "COVID-19 is a scheme for rich people and big companies to make money off of the testing and vaccines;" and "The COVID-19 vaccines are coming out so fast because they have not been carefully tested." In terms of total number of statements endorsed, Poisson regression revealed significant difference by CLI status in both bivariate and multivariable analyses (see bottom of Table 2).

## COVID-19 information sources

Table 3 presents descriptive data on COVID-19 information sources by CLI status. For CLI participants, the top five information sources were broadcast TV (56.8%), cable TV (47.9%),

**Table 3. COVID-19 information sources used last month.**

|  | Weighted % non-CLI | Weighted % CLI | p |
|---|---|---|---|
| Government and science (> = 1) | 49.8 | 39.9 | .017 |
| State or local government | 38.9 | 28.4 | .010 |
| Federal government | 33.7 | 27.2 | .110 |
| Scientific journal | 11.4 | 8.3 | .286 |
| Mainstream/news media (> = 1) | 77.6 | 81.5 | .256 |
| Local print | 18.2 | 16.1 | .586 |
| National print | 12.8 | 10.7 | .528 |
| Broadcast TV | 47.2 | 56.8 | .024 |
| Cable TV | 39.6 | 47.9 | .042 |
| Radio news | 18.1 | 16.1 | .586 |
| Online news | 31.5 | 30.8 | .929 |
| Social media | 27.1 | 25.6 | .708 |
| Social media | 27.1 | 25.6 | .708 |
| Personal sources (> = 1) | 47.5 | 35.1 | .003 |
| Employer | 16.5 | 11.8 | .138 |
| Personal networks | 40.0 | 31.0 | .026 |
| Community sources (> = 1) | 11.5 | 9.5 | .447 |
| Church | 5.1 | 3.6 | .559 |
| Community Organization | 3.9 | 4.7 | .672 |
| Other | 4.6 | 5.9 | .434 |

CLI = criminal legal involved.

personal networks (31.0%), online news (30.8%), and state or local government (28.4%). For non-CLI participants, the top five were broadcast TV (47.2%), personal networks (40%), cable TV (39.6%), State or local government (38.9%), and federal government (33.7%). For broadcast TV (p = .024), cable TV (p = .042), personal networks (p = .026), and state and local government (p = .010), the use rates between CLI and non-CLI groups were significantly different. CLI participants reported greater use of broadcast and cable TV but less use of state or local government and personal networks than non-CLI participants.

After grouping these sources into categories, two categories showed significant differences by CLI status. CLI participants reported less use of government and scientific sources (p = .017) and less use of personal sources (p = .003) than non-CLI participants. The two groups did not differ significantly on their use of mainstream and news media (p = .256), social media (p = .708), and community sources (p = .447).

## Misinformation endorsement and information sources

To answer RQ2, Table 4 presents the results of the Poisson regression models predicting total number of misinformation statements endorsed in the full sample using CLI, source usage, and their interactions (one at a time) as key predictors while controlling for sociodemographic and political background variables. CLI emerged a positive predictor of misinformation endorsement for participants not using the sources involved in the interaction in each of the five models (all IRR > 1.210, all p < .006). Additionally, two of the interactions were significant, CLI with mainstream/news media use (IRR = .785, p < .001) and CLI with social media use (IRR = .713, p = .031). In both cases, greater usage of the sources was associated with less discrepancy in misinformation endorsement between CLI and non-CLI participants. From a different angle, the interaction also showed that the association between source usage and

**Table 4. Interactions between CLI status and sources of COVID-19 information in predicting total number of misinformation statements endorsed (Full sample analysis).**

| | | 95% CI | | |
|---|---|---|---|---|
| | **IRR** | **Lower Bound** | **Upper Bound** | **p** |
| CLI | 1.233 | 1.066 | 1.425 | 0.005 |
| Government and science | 0.877 | 0.813 | 0.946 | < .001 |
| Interaction | 0.92 | 0.779 | 1.086 | 0.324 |
| CLI | 1.21 | 1.056 | 1.386 | 0.006 |
| Mainstream/News | 1.059 | 1.009 | 1.112 | 0.02 |
| Interaction | 0.785 | 0.707 | 0.871 | < .001 |
| CLI | 1.266 | 1.108 | 1.447 | < .001 |
| Social media | 1.063 | 0.929 | 1.216 | 0.374 |
| Interaction | 0.713 | 0.524 | 0.97 | 0.031 |
| CLI | 1.22 | 1.06 | 1.404 | 0.006 |
| Personal | 0.935 | 0.847 | 1.032 | 0.18 |
| Interaction | 0.802 | 0.643 | 1.001 | 0.051 |
| CLI | 1.282 | 1.123 | 1.464 | < .001 |
| Community | 1.278 | 1.112 | 1.468 | < .001 |
| Interaction | 1.038 | 0.799 | 1.348 | 0.779 |

CLI = criminal legal involved. IRR = incident rate ratio. CI = confidence interval. Each model included one CLI by source interaction, with other source usage variables controlled for. Additional control variables included gender, age, race/ethnicity, education, income, marital status, employment status, and political party/leaning. Control variables in the models are not reported.

misinformation endorsement was weaker or more negative among CLI participants than among non-CLI participants.

Table 5 presents separate Poisson models for CLI and non-CLI subsamples to further examine the relationship between source usage and misinformation endorsement within each group. In the CLI model, use of mainstream/news media (IRR = .844, p = .002) was negatively associated with, and use of community sources (IRR = 1.569, p = .002) was positively associated with misinformation endorsement. In the non-CLI model, use of government and scientific sources (IRR = .841, p < .001) were negatively associated with, and use of community sources (IRR = 1.249, p = .002) were positively associated with misinformation endorsement. Use of social media or personal sources were unrelated to misinformation endorsement for either group (all p > .105). Use of government and science sources was not associated with misinformation endorsement for the CLI group (p = .619) and use of mainstream/news media was not associated with misinformation endorsement for the non-CLI group (p = .085).

Several covariates also emerged significant in the analyses. In the CLI model, misinformation endorsement was negatively associated with older age (IRR = .712, p < .001) and positively associated with being non-Hispanic Black (IRR = 1.935, p = .003) and leaning Republican in political orientation (IRR = 1.341, p < .001). In the non-CLI model, negative covariates of misinformation endorsement included older age (IRR = .890, p < .001) and higher education (IRR = .847, p < .001), and positive covariates included being non-Hispanic Black (IRR = 2.277, p < .001), being Hispanic (IRR = 1.609, p < .001), and leaning Republican in political orientation (IRR = 1.142, p < .001).

## Discussion

This study examined the prevalence of misinformation and its relationship with COVID-19 information sources among CLI vs. non-CLI individuals in the U.S. Our nationally

**Table 5. Associations between total number of misinformation statements endorsed and sources of COVID-19 information for CLI and Non-CLI subsamples.**

| | Non-CLI | | | | CLI | | | |
| | | 95% CI | | | | 95% CI | | |
| | IRR | Lower Bound | Upper Bound | p | IRR | Lower Bound | Upper Bound | p |
|---|---|---|---|---|---|---|---|---|
| Male | 0.894 | 0.795 | 1.006 | 0.064 | 0.891 | 0.688 | 1.155 | 0.384 |
| Older age | 0.89 | 0.837 | 0.947 | < .001 | 0.712 | 0.621 | 0.816 | < .001 |
| Education | 0.847 | 0.8 | 0.898 | < .001 | 0.87 | 0.756 | 1.001 | 0.051 |
| Income | 0.991 | 0.93 | 1.056 | 0.787 | 0.98 | 0.835 | 1.149 | 0.8 |
| NH-Black | 2.277 | 1.885 | 2.751 | < .001 | 1.935 | 1.261 | 2.971 | 0.003 |
| Hispanic | 1.609 | 1.382 | 1.872 | < .001 | 0.966 | 0.682 | 1.369 | 0.846 |
| Other race/ethnicity | 1.211 | 0.956 | 1.533 | 0.112 | 1.213 | 0.769 | 1.913 | 0.406 |
| Leaning Republican | 1.142 | 1.097 | 1.19 | < .001 | 1.341 | 1.212 | 1.484 | < .001 |
| Married | 0.926 | 0.81 | 1.059 | 0.261 | 1.22 | 0.916 | 1.627 | 0.174 |
| Employed | 1.035 | 0.909 | 1.179 | 0.601 | 1.021 | 0.749 | 1.392 | 0.895 |
| Government and science | 0.841 | 0.777 | 0.909 | < .001 | 0.957 | 0.805 | 1.138 | 0.619 |
| Mainstream/news | 1.045 | 0.994 | 1.099 | 0.085 | 0.844 | 0.757 | 0.942 | 0.002 |
| Social media | 1.045 | 0.912 | 1.198 | 0.528 | 0.772 | 0.535 | 1.114 | 0.167 |
| Personal | 0.931 | 0.841 | 1.03 | 0.163 | 0.813 | 0.632 | 1.045 | 0.105 |
| Community | 1.249 | 1.085 | 1.438 | 0.002 | 1.569 | 1.184 | 2.08 | 0.002 |
| | LR Chi-Square = 298.35, df = 15, p < .001 | | | | LR Chi-Square = 106.91, df = 15, p < .001 | | | |

CLI = criminal legal involved. IRR = incident rate ratio. CI = confidence interval. NH = non-Hispanic. Adj. = adjusted. LR = likelihood ratio.

representative survey data showed that the rate of misinformation endorsement was higher among CLI respondents than among their non-CLI counterparts. It appears clear that the CLI population is indeed more prone to favoring COVID-19 misinformation than is the non-CLI population in the U.S.

An important caveat for these results is that the misinformation items used in this study were based on earlier qualitative research with CLI participants [64]. It is thus possible that this specific set of items might have an inherent bias toward greater endorsement among CLI respondents. While this is a reasonable possibility, it is notable that most, if not all, of the misinformation tested in this study is also widely circulated in the general population. For example, the two statements with the highest rates of endorsement in our data were about rushed vaccine development and contracting COVID-19 through testing. These misperceptions are widely documented in previous general population studies and continuing misinformation surveillance efforts [3, 41, 44]. We saw substantial gaps between the CLI and non-CLI groups on these two items, with the one about vaccine development reaching almost 20%. This suggests that the greater prevalence of misinformation among the CLI population is a real and broad phenomenon that is not likely to be entirely driven by the specific set of items used in this study.

On the other hand, it should be noted that the difference between CLI and non-CLI groups on some of the misinformation items was not striking. Indeed, most misinformation items exhibited relatively modest levels of endorsement in both groups. The levels of endorsement observed in this study, however, were generally comparable to those reported in other national studies. For example, the COVID States Project assessed a number of vaccine-related misinformation beliefs in January 2022 and the endorsement rates ranged from 5% to 10% [3]. With the potential harms of misinformation in mind, we believe that even low levels of prevalence and relatively small differences between groups deserve careful research and policy attention.

Previous research, both before and during the pandemic, suggests that health knowledge and beliefs, misperceptions included, are intricately associated with the sources people use to obtain health information [3, 48, 49]. Our data showed different patterns of information source usage among CLI and non-CLI individuals. CLI individuals were significantly less reliant on government/scientific and personal sources, and more reliant on broadcast and cable TV, for COVID-19 information compared to their non-CLI peers.

A few patterns are notable in the relationship between source usage and misinformation endorsement for CLI and non-CLI respondents. First, use of government and scientific sources was associated with reduced misinformation endorsement among non-CLI individuals, but not among CLI individuals. The IRR of government/scientific sources for the non-CLI subsample was well below 1 (.841 to be exact), indicating a negative association, while the IRR for the CLI subsample was virtually indistinguishable from 1 (.957 to be exact), indicating no relationship. Second, mainstream and news media use was negatively associated with misinformation endorsement for the CLI subsample. For the non-CLI subsample, the association was not significant. The difference between the CLI and non-CLI subsamples was significant by virtue of the interaction analysis. Third, the use of social media was not significantly associated with misinformation endorsement in either subsample. However, the interaction between social media use and CLI status was significant, and it appears that this interaction was mostly driven by a negative association in the CLI subsample. Fourth, use of community sources was positively associated with misinformation endorsement in both subsamples, suggesting potential risk of gaining COVID-19 information from these sources for both CLI and non-CLI individuals.

Among the sociodemographic and political background variables, we saw a similar pattern between the CLI and non-CLI groups. Younger age, lower education, being non-Hispanic Black (vs. White), and leaning Republican in political orientation were associated with greater misinformation endorsement for both groups. The only difference is that Hispanic/Latinx/e individuals were more likely than White individuals to endorse misinformation among non-CLI individuals, whereas among CLI individuals there was no difference between the two racial/ethnic groups.

This study is the first to examine the prevalence of COVID-19 misinformation and its relationship with information source usage among the CLI population, identified through a nationally representative sample. Our results indicate that CLI communities are indeed more vulnerable to COVID-19 misinformation compared to non-CLI segments of the general population. Although not all misinformation is equally detrimental [69], the nature of the misinformation assessed in the current study suggests a real possibility of deleterious consequences. For example, rushed vaccine development was the most widely endorsed misinformation in this study. This is a common myth about the COVID-19 vaccines [42] and has been linked to vaccine hesitancy in both observational and experimental studies [2, 3]. Clearly, misinformation such as this can have a disproportional impact on CLI communities, adding to their already heavy burden of health disparities during the pandemic. How to protect the CLI population from the influence of detrimental misinformation is a question of significance and urgency.

Our data on information sources provide useful insights into both the dissemination pathways of misinformation and potential channels to use for remediation. First, we note that government and scientific sources are not only underutilized by CLI individuals, but also unhelpful in the mitigation of misinformation within this population. We suspect that this dual deficit has much to do with the deep-rooted distrust in government and medical establishment among CLI communities. Indeed, one of the most pronounced themes in the existing literature on CLI-related health disparities [46, 58, 62, 63], as well as in our own qualitative

research [64], is the lack of trust in information disseminated by government agencies and public health institutions. Thus, even though government and public health messaging is perhaps less likely to be influenced by misinformation, it often fails to reach and engage the CLI population. As a result, government and scientific sources may be marginalized in the fight against misinformation in CLI communities.

Mainstream and news media appear to have played a protective role against misinformation in the CLI population. This finding is consistent with some previous research that showed exposure to traditional media to be associated with lower beliefs in conspiracy theories and misinformation [53]. Broadcast and cable TV, in particular, are widely used by CLI individuals for COVID-19 information. Although the content featured in these media outlets is necessarily complex and likely influenced by the political leaning of specific channels, the fact that they have significant reach and a negative association with misinformation endorsement among CLI communities suggests that they may be a productive venue for both the dissemination of accurate information and strategic messaging aimed at misinformation correction. Social media, on the other hand, have no clear relationship with misinformation for either the CLI or non-CLI population. These findings are interesting because social media are often blamed for the spread of misinformation during the pandemic [44, 70, 71]. Our data suggest that broad conclusions may be premature without diving more deeply into the dynamic, countervailing influences of factual versus false information on these vast platforms [72].

A positive association between the use of community sources and misinformation endorsement appear to exist in both CLI and non-CLI communities. Although these sources are used by a relatively low number of community members, they include important social institutions such as community centers and churches. Our findings urge attention to the nature of the COVID-19 information that flows through these social hubs and greater efforts to monitor and mitigate misinformation in these community settings. Strategic partnerships with community leaders and institutions could be useful to these efforts.

This study fills an important gap in the existing literature by bringing together two hitherto independent lines of research, one concerning the well-being of the CLI population and the other the rising tides of health misinformation during the pandemic. Its scientific contributions are two-fold. First, this study has provided much needed evidence to illuminate the heightened vulnerability to misinformation among a uniquely marginalized population. Second, it paves the way for future, theory-driven research to investigate the intersection between CLI and misinformation in a systematic manner. As a unique social determinant of health [6], CLI is an important cue of social identity that may influence both selective exposure to and the processing of health (mis)information. Distrust in and resistance to government sources of information, for example, may represent strong outgroup bias [73] that can feed into a self-reinforcing belief echo [74]. Interesting hypotheses such as these can be tested to advance our understanding of the acquisition and persistence of misconceptions among CLI communities.

This study has important implications for protecting and enhancing the well-being of the CLI population during the pandemic. First, misinformation is an important risk factor that may exacerbate the health disparities faced by this population. Targeted efforts are needed to monitor and combat the spread and influence of misinformation in CLI communities. Second, mainstream media, particularly television, are important channels to reach and engage the CLI population. While comprehensive public health messaging may seek to leverage all available media, purposeful placement of messages in traditional media may prove especially beneficial for members of CLI communities. Third, efforts are needed to promote the use of government and scientific sources among the CLI population. A focal point in these efforts will be to build trust through messages that acknowledge past and ongoing mistreatments, validate feelings of neglect, mistrust, and betrayal, and express willingness to work hard to regain the CLI

communities' confidence in government agencies and public health institutions. Fourth, and related to the point above, public health communication should identify sources that already enjoy relatively high levels of trust among CLI communities and partner with these sources as a conduit for the dissemination of accurate COVID-19 information. Fifth, care needs to be taken when engaging community sources. Simply pushing accurate information into these environments may cause confusion and uncertainty as communities work to make sense of new information. Close monitoring and effective mitigation of misinformation in community settings is an important task in its own right. It is also essential for the success of broader pandemic-related health promotion efforts. Finally, policy makers should continue to look for and implement effective strategies to contain and counter health misinformation in the media and social environments. To be sure, there is no simple solution to the problem. The COVID-19 pandemic has spurred numerous laws around the globe to combat misinformation, but their effectiveness (and for some, their legitimacy) remains in question [75]. In a democratic society, it appears that a reasonable path forward should involve concerted policy efforts to promote public awareness, media literacy, technological interventions, and legal accountability, among other initiatives.

This study has several limitations. First, the list of misinformation, while carefully extracted from previous qualitative research, is necessarily time-bound, thus merely a snapshot of reality in a quickly evolving pandemic. Second, the information sources surveyed in this study did not capture the political leaning of specific outlets within the same media (e.g., Fox news versus MSNBC). This has limited the scope of our investigation. Third, source usage was measured as a simple dichotomy in this study. The amount of usage of each source was thus unclear. Fourth, the cross-sectional nature of the current data does not support causal interpretation of the relationship between source usage and misinformation endorsement. Fifth, the study measured criminal legal system involvement as a dichotomous variable. Important differences might exist among the CLI individuals depending on the amount and type of involvement with the legal system and time since involvement. Sixth, as seen in other studies reliant on household surveys, we had a modest response rate. Nevertheless, the AmeriSpeak panel's response rate of 37% is one of the highest for comparable national probability-based household panels [76]. We weighted the data to national census benchmarks, taking into account selection probabilities and addressed possible non-response bias with statistical weights and non-response adjustments. Finally, despite a fairly large national sample, the number of CLI respondents in this study was relatively small, resulting in limited power in some of our analyses. The unequal sample sizes of CLI versus non-CLI respondents could also have affected the precision and stability of some of the estimates. Future research should consider oversampling CLI individuals to address these issues. These limitations notwithstanding, this study contributes useful evidence to identify misinformation as an important concern for the well-being of CLI individuals during the pandemic. Efforts to contain and mitigate misinformation in this vulnerable population using appropriate information sources are warranted.

## Supporting information

**S1 Data.**
(SAV)

## Author Contributions

**Conceptualization:** Xiaoquan Zhao, Faye S. Taxman.

**Data curation:** Bruce G. Taylor, Phoebe A. Lamuda, Harold A. Pollack, John A. Schneider.

**Formal analysis:** Xiaoquan Zhao.

**Funding acquisition:** Bruce G. Taylor, Harold A. Pollack, John A. Schneider, Faye S. Taxman.

**Investigation:** Aayushi Hingle, Cameron C. Shaw, Amy Murphy, Breonna R. Riddick, Rochelle R. Davidson Mhonde.

**Methodology:** Xiaoquan Zhao, Bruce G. Taylor, Phoebe A. Lamuda, Harold A. Pollack, John A. Schneider, Faye S. Taxman.

**Supervision:** Faye S. Taxman.

**Writing – original draft:** Xiaoquan Zhao, Aayushi Hingle, Cameron C. Shaw, Amy Murphy, Breonna R. Riddick.

**Writing – review & editing:** Rochelle R. Davidson Mhonde, Bruce G. Taylor, Phoebe A. Lamuda, Harold A. Pollack, John A. Schneider, Faye S. Taxman.

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
