## [Decision Letter · Decision Letter 0]

12 Apr 2023

PONE-D-22-16444COVID misinformation endorsement among Americans who are criminal legal involved: Prevalence and relationship with information sourcesPLOS ONE

Dear Dr. Zhao,

Thank you for submitting your manuscript to PLOS ONE. After careful consideration, we feel that it has merit but does not fully meet PLOS ONE’s publication criteria as it currently stands. Therefore, we invite you to submit a revised version of the manuscript that addresses the points raised during the review process.

Please address all comments by reviewers. In addition, substantive ideas for policies to combat misinformation was mentioned as a need in private comments by one of the reviewers. 

We look forward to receiving your revised manuscript.

Kind regards,

Janet E Rosenbaum, Ph.D.

Academic Editor

PLOS ONE

Journal Requirements:

2. In the ethics statement in the Methods, you have specified that verbal or online consent was obtained. Please provide additional details regarding how this consent was documented and witnessed, and state whether this was approved by the IRB.

“FST (#U2CDA050097), JS (1U01MD017414-01), and BT, PAL, HP, and JS (# U2CDA050098) were supported by the U.S. National Institute on Drug Abuse (https://nida.nih.gov/). The funder had no role in study design, data collection and analysis, decision to publish, or preparation of the manuscript.”

Reviewers' comments:

Reviewer's Responses to Questions

**Comments to the Author**

1. Is the manuscript technically sound, and do the data support the conclusions?

Reviewer #1: Yes

Reviewer #2: Partly

Reviewer #3: Yes

2. Has the statistical analysis been performed appropriately and rigorously? 

Reviewer #1: Yes

Reviewer #2: No

Reviewer #3: I Don't Know

3. Have the authors made all data underlying the findings in their manuscript fully available?

Reviewer #1: No

Reviewer #2: Yes

Reviewer #3: Yes

4. Is the manuscript presented in an intelligible fashion and written in standard English?

Reviewer #1: Yes

Reviewer #2: Yes

Reviewer #3: Yes

5. Review Comments to the Author

Reviewer #1: This paper investigates the differences between a sample of Americans and a subsample of individuals with prior history of criminal legal system involvement (CLI) regarding their endorsement of COVID-19 misinformation and its relation to media use patterns. Using survey questionnaires and Poisson regression analyses, the authors find a number of differences, and also some similarities, between the two groups. Most notably, CLI persons show a greater frequency of misinformation endorsement than the general populace. Furthermore, some media use types were related to endorsement.

This is an interesting study, the paper is clearly written, and the interpretation of the results is generally warranted. I can see this study offering a valuable contribution to the literature on misinformation endorsement, especially by providing a more detailed look at how membership of specific social subgroups may affect how media are used and how this is connected to developing misperceptions. Before publication, however, I have a few questions and suggestions, which I will list below. Some related to how the paper is written, others are related to the analyses and their interpretations.

Disclaimer: Although I am a social scientist, and the methodology presented here is largely familiar to me, I cannot claim to be well-versed in the specific analyses strategies used here, especially the Poisson regression.

COMMENTS:

1. One of my chief concerns is to do with the data, more specifically the extremely different sample sizes of the main and subsample, with the general sample (N=1161) forming around 9-10 times the size of the subsample (N=168). This would increase the chances of (very) unequal variances, and associated problems with over or underestimating effect p-values. I wonder what you think of this matter, and would suggest that you address it in the paper, by offering a bit more elaborate explanation of the decisions involved in this regard, as well as informing the reader about the variances in the two samples.

2. Relatedly, as you rightly acknowledge in the discussion section (p. 29), because of the small CLI sample, there are some issue with the power of the analyses. Could you expand a little more on that in the paper? I.e., while you acknowledge the problem, you do not really offer a comforting rebuttal. Moreover, there is no information about a priori or post hoc power analyses performed. One thing is that one could argue that the analyses should be powered on the smallest sample, i.e. N=168. Would this have consequences for your results and their interpretation, also with regards to the fairly modest effect sizes (see below)?

3. Regarding the interpretations of the results, I noticed that there are many instances of the “there was a difference/correlation between x and y, but it was not significant” variant. I would suggest that it would be more correct to state that when you did not find a significant difference/correlation, there is no difference/correlation, without qualification.

4. In my opinion, the fact that most effect sizes are in the small-moderate range, deserves more attention. I.e., there are some differences between the two groups, but the differences are perhaps not that dramatic. Also considering the fact that most misinformation statements exhibit fairly low endorsement

5. Can you give us a little more information on how the subsample is or isn’t representative of the larger CLI population?

6. Was there any factor analysis done on the misinfo statements? I ask partly because one of the items seems a bit of an outlier, representing more of a conspiracy thinking than purely misconceptions: “COVID-19 is a scheme for rich people and big companies to make money off of the testing and vaccines.”

7. The introduction to the paper is very thorough (p3-8). In fact, it is a little overwhelming in its detail. I would say that the paper would be improved if the introduction would be a little more concise, and many of the arguments made would be equally strong if the data and findings from prior research could be presented in a more summarized form. I for me found it sometimes difficult to stay focused through the multitude of percentages presented.

8. Likewise, in my opinion there is no need to repeat all the findings in the discussion, because we can all read it in the results section. Just a quick summary of the most important results, and then on to the interpretation of the results.

9. This would also free up some room for two of my other suggestions. Firstly, I would like to hear a bit more about the scientific contribution or relevance of focusing on this particular group. As it stands, the focus is much more on the societal relevance. Secondly, I wonder if the paper and its scientific contribution would not be stronger if you tried to find a theoretical framework, from which you could derive testable hypotheses? Now, the paper leaves the reader with the question of what it all means. This could be addressed both in the introduction and discussion section, the latter of which could use some more engagement with prior research, and especially extant theoretical notions in the field.

Minor things:

1. Table 1: for the reader’s convenience, could you provide info on the meaning of the p-values, either in the column or below the table in a note?

2. In Table 5 it says ‘oder age’, which should be ‘older’ I guess. But reading the text, I got confused, because there It says that younger age associated with misinformation endorsement (p23). This does not seem to correspond with what’s in the table for older age. Or did I misread?

Thanks for allowing me to read this paper, and the best of luck with it!

Reviewer #2: The article investigates the prevalence of COVID-19 related misinformation and its relationship with COVID-19 information sources used among Americans experiencing CLI. The study surveyed a nationally representative sample of American adults aged 18+ including a subsample of CLI individuals. The results show that CLI participants endorsed a greater number of misinformation statements and reported less use of government and scientific sources and personal sources for COVID-19 information compared to non-CLI participants. The study suggests that building trust in important information sources is critical to the containment and mitigation of COVID-related misinformation in the CLI population.

Although the authors claimed that this study is the first to focus on individuals experiencing CLI, the findings fail to offer any significant new insights into the existing knowledge in the field. Despite using a nationally representative sample, the sample size for the CLI population may be considered too small to generate reliable and robust results. Additionally, the use of Poisson regression models in the analysis may oversimplify the complex ways in which misinformation can impact individuals and fails to fully consider the contextual factors that contribute to its spread.

The authors need to revise the literature review especially in the "CLI as social determinants of health" subsection to ensure that it is more relevant and directly tied to the goals and objectives of the study. The literature review should clearly and effectively support the authors' argument and provide a comprehensive understanding of the current knowledge and research on CLI as a social determinant of health. Additionally, the literature review should highlight the gap in the existing knowledge that the authors aim to address through their study.

Although the authors mentioned that the data is available, they did not provide the link to the platform where the data underlying the findings in the manuscript can be accessed.

Reviewer #3: The paper was written in clear and simple language, which will benefit the non-scientific/academic reader. While a few sentences and clauses were unclear and perhaps could be revised to provide a much clearer explanation, I found the paper acceptable. I am not an expert in the quantitative methodology, therefore will not be able to comment on the method and analysis of the study.

6. PLOS authors have the option to publish the peer review history of their article (what does this mean?). If published, this will include your full peer review and any attached files.

Reviewer #1: **Yes: **Gabi Schaap

Reviewer #2: No

Reviewer #3: **Yes: **Suffian Hadi Ayub

---

## [Author Response · Author response to Decision Letter 0]

2 Jul 2023

A detailed response to reviewers has been uploaded. The additional information requested in the editor's decision letter has been provided in the cover letter.

---

## [Decision Letter · Decision Letter 1]

21 Nov 2023

PONE-D-22-16444R1COVID misinformation endorsement among Americans who are criminal legal involved: Prevalence and relationship with information sourcesPLOS ONE

Dear Dr. Xiaoquan Zhao,

Thank you for submitting your manuscript to PLOS ONE. After careful consideration, we feel that it has merit but does not fully meet PLOS ONE’s publication criteria as it currently stands. Therefore, we invite you to submit a revised version of the manuscript that addresses the points raised during the review process.

The submitted manuscript has undergone a rigorous evaluation process carried out by the assigned reviewers. As a result of this evaluation, several issues have been identified and duly communicated. In light of these observations, I kindly urge you to promptly address these concerns by making the necessary corrections in order to enhance the quality of your article. Additionally, I kindly request that you submit the revised final version through the designated system as soon as possible. The submitted manuscript has undergone a rigorous evaluation process carried out by the assigned reviewers. As a result of this evaluation, several issues have been identified and duly communicated. In light of these observations, I kindly urge you to promptly address these concerns by making the necessary corrections in order to enhance the quality of your article. Additionally, I kindly request that you submit the revised final version through the designated system as soon as possible. The submitted manuscript has undergone a rigorous evaluation process carried out by the assigned reviewers. As a result of this evaluation, several issues have been identified and duly communicated. In light of these observations, I kindly urge you to promptly address these concerns by making the necessary corrections in order to enhance the quality of your article. Additionally, I kindly request that you submit the revised final version through the designated system as soon as possible.

We look forward to receiving your revised manuscript.

Kind regards,

Shahabedin Rahmatizadeh, Ph.D.

Academic Editor

PLOS ONE

Journal Requirements:

Reviewers' comments:

Reviewer's Responses to Questions

**Comments to the Author**

1. If the authors have adequately addressed your comments raised in a previous round of review and you feel that this manuscript is now acceptable for publication, you may indicate that here to bypass the “Comments to the Author” section, enter your conflict of interest statement in the “Confidential to Editor” section, and submit your "Accept" recommendation.

Reviewer #1: (No Response)

Reviewer #4: All comments have been addressed

Reviewer #5: (No Response)

2. Is the manuscript technically sound, and do the data support the conclusions?

Reviewer #1: Yes

Reviewer #4: Yes

Reviewer #5: Yes

3. Has the statistical analysis been performed appropriately and rigorously? 

Reviewer #1: Yes

Reviewer #4: Yes

Reviewer #5: Yes

4. Have the authors made all data underlying the findings in their manuscript fully available?

Reviewer #1: Yes

Reviewer #4: Yes

Reviewer #5: (No Response)

5. Is the manuscript presented in an intelligible fashion and written in standard English?

Reviewer #1: Yes

Reviewer #4: Yes

Reviewer #5: Yes

6. Review Comments to the Author

Reviewer #1: I find the authors have addressed my comments adequately. I have two remaining suggestions:

1. Re. comment 2 on the power issues: your addition to the analyses section is sufficient, however, I would recommend you add 1-2 references to back up your statement about the cases per variable ratio

2. Re comment 5: I recommend incorporating your response (perhaps slightly abbreviated) into the sample characteristics paragraph, or alternatively into an endnote, as I do not think this information is directly apparent to the uninitiated

Reviewer #4: The research is significant due to numerous distinct factors: The research contributes to understanding misinformation as a form of disparity affecting CLI communities during the pandemic. It specifically targets criminally legal-involved (CLI) communities, a group often overlooked in misinformation research.

Additionally, the study examines both the endorsement of COVID-19 misinformation and the relationship with various information sources in CLI communities.

Finally, the paper employs detailed logistic and Poisson regression analyses, enhancing the reliability and depth of the findings with a detailed misinformation analysis.

Reviewer #5: Please include the -19 in the COVID to be more specific.

Please consider improving your entire topic also to make it clearer.

i.e.: Examining the Endorsement of COVID-19 Misinformation Among Individuals in the Criminal Legal System:….

Specified each of the questions the research seeks to answer immediately after the background and the objective. This can be according to the group of analysis in the study.

7. PLOS authors have the option to publish the peer review history of their article (what does this mean?). If published, this will include your full peer review and any attached files.

Reviewer #1: **Yes: **Gabi Schaap

Reviewer #4: No

Reviewer #5: **Yes: **Mahfooz Ahmed

---

## [Author Response · Author response to Decision Letter 1]

21 Nov 2023

A response to reviewer has been uploaded with the resubmission.

---

## [Decision Letter · Decision Letter 2]

18 Dec 2023

Endorsement of COVID-19 misinformation among criminal legal involved individuals in the United States: Prevalence and relationship with information sources

PONE-D-22-16444R2

Dear Dr. Xiaoquan Zhao

We’re pleased to inform you that your manuscript has been judged scientifically suitable for publication and will be formally accepted for publication once it meets all outstanding technical requirements.

Kind regards,

Shahabedin Rahmatizadeh, Ph.D.

Academic Editor

PLOS ONE

Additional Editor Comments (optional):

Reviewers' comments:

Reviewer's Responses to Questions

**Comments to the Author**

1. If the authors have adequately addressed your comments raised in a previous round of review and you feel that this manuscript is now acceptable for publication, you may indicate that here to bypass the “Comments to the Author” section, enter your conflict of interest statement in the “Confidential to Editor” section, and submit your "Accept" recommendation.

Reviewer #6: All comments have been addressed

2. Is the manuscript technically sound, and do the data support the conclusions?

Reviewer #6: Yes

3. Has the statistical analysis been performed appropriately and rigorously? 

Reviewer #6: Yes

4. Have the authors made all data underlying the findings in their manuscript fully available?

Reviewer #6: Yes

5. Is the manuscript presented in an intelligible fashion and written in standard English?

Reviewer #6: Yes

6. Review Comments to the Author

Reviewer #6: The research investigates the prevalence of COVID-19-related misinformation among individuals involved in the criminal legal system (CLI) in the United States. CLI participants endorsed a greater number of misinformation statements compared to non-CLI participants.

Statistical analyses are described in sufficient detail.

Results and discussion are presented in an appropriate fashion and are supported by the data.

The study contributes useful evidence to identify misinformation as an important concern for the well-being of CLI individuals during the pandemic.

7. PLOS authors have the option to publish the peer review history of their article (what does this mean?). If published, this will include your full peer review and any attached files.

Reviewer #6: No

---

## [Editor Report · Acceptance letter]

29 Dec 2023

PONE-D-22-16444R2 

PLOS ONE

Dear Dr. Zhao, 

I'm pleased to inform you that your manuscript has been deemed suitable for publication in PLOS ONE. Congratulations! Your manuscript is now being handed over to our production team.

Kind regards, 

on behalf of

Dr. Shahabedin Rahmatizadeh 

Academic Editor

PLOS ONE